# Comparison between Two Adaptive Optics Methods for Imaging of Individual Retinal Pigmented Epithelial Cells

**DOI:** 10.3390/diagnostics14070768

**Published:** 2024-04-04

**Authors:** Elena Gofas-Salas, Daniel M. W. Lee, Christophe Rondeau, Kate Grieve, Ethan A. Rossi, Michel Paques, Kiyoko Gocho

**Affiliations:** 1Department of Photonics, Institut de la Vision, INSERM, CNRS, Sorbonne Université, 17 rue Moreau, F-75012 Paris, France; elena.gofas@inserm.fr; 2CIC 1423, CHNO des Quinze-Vingts, INSERM-DGOS 28 rue de Charenton, F-75012 Paris, France; mpaques@15-20.fr (M.P.); kknaka17@gmail.com (K.G.); 3Department of Bioengineering, University of Pittsburgh, Pittsburgh, PA 15213, USA; dal208@pitt.edu (D.M.W.L.); rossiea@pitt.edu (E.A.R.); 4Imagine Eyes, F-91400 Orsay, France; crondeau@imagine-eyes.com; 5Department of Ophthalmology, University of Pittsburgh School of Medicine, Pittsburgh, PA 15260, USA

**Keywords:** retinal pigment epithelium, adaptive optics, autofluorescence imaging, transscleral illumination

## Abstract

The Retinal Pigment Epithelium (RPE) plays a prominent role in diseases such as age-related macular degeneration, but imaging individual RPE cells is challenging due to their high absorption and low autofluorescence emission. The RPE lies beneath the highly reflective photoreceptor layer (PR) and contains absorptive pigments, preventing direct backscattered light detection when the PR layer is intact. Here, we used near-infrared autofluorescence adaptive optics scanning laser ophthalmoscopy (NIRAF AOSLO) and transscleral flood imaging (TFI) in the same healthy eyes to cross-validate these approaches. Both methods revealed a consistent RPE mosaic pattern and appeared to reflect a distribution of fluorophores consistent with findings from histological studies. Interestingly, even in apparently healthy RPE, we observed dynamic changes over months, suggesting ongoing cellular activity or alterations in fluorophore distribution. These findings emphasize the value of NIRAF AOSLO and TFI in understanding RPE morphology and dynamics.

## 1. Introduction

Retinal diseases vary widely, but most of them cause visual symptoms, often leading to blindness [1], one of the biggest fears of today’s world population [2]. Some retinal disorders, such as age-related macular degeneration (AMD) or inherited retinal diseases (IRDs), including retinitis pigmentosa (RP), remain poorly understood and without efficient treatment. In the last decade, novel stem cell therapies have been developed to address these pathologies, and a few clinical trials on humans are currently underway [3]. However, one of the most significant challenges in the current development of these therapies is to reach the capacity to monitor the structural and functional changes in the retina driven by these treatments at a cellular level. In particular, it is known that RPE cells play an important role in the progression of these pathologies, though the mechanisms involved are not yet well understood. It is, therefore, crucial to study alterations in the RPE to assist the development of effective treatment methods for AMD and IRDs.

Although there are a few existing clinical modalities to detect and image the RPE, they only allow an assessment of the state of the layer at the tissue level, and subtle alterations cannot be appreciated using existing clinical tools. Optical coherence tomography (OCT) generates cross-sections of the retina and allows clinicians to observe the RPE layer and follow significant changes over time [4]. The most commonly used modality to evaluate the RPE state in the clinic is short and near-infrared wavelength autofluorescence scanning laser ophthalmoscope (SLO), which, through excitation of endogenous fluorophores [5] allows the user to observe damage to the RPE [6]. To observe fine cellular changes, various teams applied Adaptive Optics (AO) technology to these modalities, first leading to cellular resolution of autofluorescence images of the RPE in the SLO camera [7,8,9,10,11,12,13] and later on revealing individual cells in OCT cross-sections [14,15,16]. Autofluorescence imaging with the AOSLO requires longer exposure times (from 30 s to 90 s) to provide high-resolution images of the RPE. Autofluorescence imaging in histology samples has enabled the study of the different types of fluorophores that are excited in these cells [17]. This has led to the hypothesis that, on top of lipofuscin, melanin pigment was also excited using infrared light [8,11,17]. *In vivo* autofluorescence AOSLO could similarly be used to facilitate the work for understanding the spectral fluorescence of these endogenous fluorophores in the living eye. In particular, near-infrared autofluorescence (NIRAF) AOSLO could potentially be used to help characterize melanin distribution in this layer, which is believed to play an important role in diseases such as AMD. Recently, another AO technique has been developed to image RPE cells through the development of transscleral imaging. The first system implemented by LaForest et al. [18,19,20] consisted of sequentially illuminating the retina through the sclera at each side of the eye pupil and then subtracting each generated image. This system, known as transscleral optical phase imaging (TOPI), revealed the RPE cell mosaic reportedly through phase contrast. In this study, we present another system using transscleral illumination, a commercial prototype named transscleral flood illumination (TFI) camera (Imagine Eyes, Orsay, France), which is currently installed at the Quinze-Vingts Hospital in Paris. Although the TFI also illuminates the retina through the sclera, it is based on a different image formation as both transscleral beams are simultaneously shone on opposite sides of the pupil to obtain an optical sum of both signals. Unlike TOPI, which uses the difference between two oblique illuminations to reduce absorption and increase phase signal, the simultaneous illumination of the TFI suggests that its image contrast is mostly derived from absorption. Both transscleral systems exploit the fact that the retina is illuminated through the sclera to avoid the strong backscattering of the photoreceptors, which masks RPE signal when detecting the light coming from the pupil in standard retinal imaging [18,19,20]. In this work, we aim to compare images of the same healthy RPE cells obtained with this new transscleral modality and with near-infrared autofluorescence (NIRAF) images from AOSLO. In particular, we intend to validate the observation of RPE cells in the TFI modality to better understand the origin of contrast in both modalities and discover potential biomarkers of healthy RPE.

## 2. Materials and Methods

### 2.1. Adaptive Optics Ophthalmoscopes

Participants were imaged with two types of ophthalmoscopes capable of revealing RPE cell mosaic: the Transscleral Flood Illumination camera (rtx1 TFI, Imagine Eyes, France) and two Near-Infrared Autofluorescence (NIRAF) AOSLO. They are described below in detail.


**Transscleral Flood Illumination (TFI) ophthalmoscope**
The TFI is an AO retinal camera with a transscleral flood illumination system (rtx1 TFI, Imagine Eyes, France). The ophthalmoscope uses two LED arrays at an 810 nm wavelength, which shine light through the sclera from both sides of the pupil (Figure 1) and generates images showing the boundaries of RPE cells as bright pixels and the centers of each cell as darker pixels. The acquisition time is 6 s per video.
**NIRAF Adaptive Optics Scanning Laser Ophthalmoscopes (AOSLO)**
Subjects 1 and 4 were imaged with the Paris near-infrared autofluorescence (NIRAF) AOSLO situated at the Quinze-Vingts Hospital, which has been previously described [11]. Subjects 2 and 3 were imaged with the NIRAF AOSLO situated at the Pittsburgh Vision Institute detailed in [21]. Both systems exploited near-infrared excitation of RPE fluorophores with light at similar wavelengths of 757 nm and 720 nm, respectively, which was used to generate images of the RPE cell mosaic in all subjects through detection of the autofluorescence. Image sequences from both systems were registered and corrected for distortions using a custom algorithm described in [22] and then averaged. Both NIRAF system characteristics can be found in Table 1.

### 2.2. Cohort Description and Image Acquisition

Four healthy volunteers (1 female, 3 males) over 18 years old were recruited. Routine fundus imaging was performed on volunteers on multiple occasions to verify that the retina was healthy and that there was no sign of retinal pathology. The study spanned 7 years; therefore, subjects have aged between the first and last image acquisitions. In Table 2, we detailed the acquisitions on each system and the corresponding subjects’ ages. In summary, NIRAF images on the AOSLO at the Quinze-Vingts in 2017 were acquired in subjects 1 and 4. NIRAF images on AOSLO at the Pittsburgh Vision Institute in 2022 were taken in subjects 2 and 3. Finally, all subjects were imaged on the TFI system in 2023, and additionally, images were acquired on subject 1 from 2021 to 2023. Dilating drops were administered to subjects 15 min prior to imaging to enlarge their pupil via one drop each of phenylephrine hydrochloride (2.5%) and tropicamide (1%). We generated image montages from 10 degrees in the temporal retina (10°T) to the fovea of subjects 1–3 in both NIRAF and TFI systems. Then, images were acquired on all subjects on a region of interest at 10°T. An image of the photoreceptor layer is simultaneously acquired in all systems, using brightfield detection in the TFI and confocal in the NIRAF systems.

The study was conducted in accordance with the Declaration of Helsinki and approved by the Ethics Committee in Paris (IDRCB number: 2019-A00942-55, CPP Sud Est III: 2019-021-B) and the Institutional Review Board of the University of Pittsburgh (CR20040340-010) for this study in humans. Written informed consent was obtained after the risks were explained to the participants both verbally and in writing. Before the start of imaging, the power of the laser was measured and recorded, ensuring the laser power was below the maximum MPE outlined by the ANSI standard guidelines.

### 2.3. Image Processing and Analysis

Background subtraction of TFI images was performed by subtracting a Gaussian-filtered version of the image. The i2k software (Imagine Eyes, Orsay, France, https://www.imagine-eyes.com/products/i2kretina, accessed on 4 October 2023) was used to assemble the montages of average images from various eccentricities taken with the TFI system, as well as to align images from the same region taken at different times. NIRAF images were montaged with a Matlab (The MathWorks, Natick, MA, USA) custom-made algorithm [23] inspired by [24]. TFI and NIRAF images of RPE cells display slightly different contrast, which renders their superposition difficult. We manually aligned the average images of photoreceptors from TFI with NIRAF confocal images and used them as a reference to manually superimpose the RPE images of both modalities. This alignment has been done using Photoshop software (version 24.2.1, Adobe, San Jose, CA, USA). The power spectrum density of the images at 10°T from both modalities were computed using custom-made Matlab software (R2022b). We then extracted cell spacing and density metrics using the estimation method described in [25] for all subjects.

## 3. Results

### 3.1. Images of TFI and NIRAF of Same Regions

Reconstructed montages of NIRAF and TFI images revealed the characteristic appearance of the RPE cell mosaic in both modalities as shown for subject #1 in Figure 2. These first TFI images of the RPE layer show dark cell centers surrounded by brighter cellular borders.

In Figure 2, various zooms of the same regions across 10° from the fovea to the temporal side show a closer look at the cell mosaic generated by each modality. Images from both modalities present strong similarities, suggesting we are observing the same cells. They both share a similar appearance of the RPE cells with a hyposignal center and a hyper-signal surround. Similarly, in both cases, the RPE mosaic is wider and clearer at further eccentricities, while close to the fovea, cell size seems to decrease, and the cell mosaic signal is not as clearly resolved.

In NIRAF imaging, we can sometimes observe photoreceptor signals in the autofluorescence image [11]. This leakage effect is most apparent in the NIRAF zoom on the right of Figure 2 corresponding to the fovea, although it can also be observed to a lesser extent on the other NIRAF zooms. Although such leakage is not as apparent in the TFI images, it remains difficult to distinguish a clear RPE mosaic, and we notice what seems to be small cells in an irregular arrangement closer to the fovea. On the other hand, the modalities display significant differences. TFI images display a higher signal contrast compared to NIRAF. Also, there are no vessel shadows in TFI images, contrary to those obtained with NIRAF. However, TFI images present a modulation of the background signal with low frequencies, which does not exist in NIRAF because the autofluorescence signal is only emitted from the RPE cells and thus provides inherent optical sectioning only to the RPE layer. Additionally, the confocal pinhole provides further filtering of any residual out-of-focus light reaching the detector. We were able to image the same retinal region at 10° from the fovea in the temporal retina (10°T) on the same subjects with both TFI and NIRAF modalities, allowing us to generate images of the same cells with these two different imaging systems. Figure 3 shows for each subject the average image in NIRAF and TFI and the superposition of the radial average of power spectrum densities (PSDs) computed on the NIRAF and TFI images.

All PSDs show a peak corresponding to the modal spacing of the RPE cells. NIRAF and TFI peaks are perfectly aligned for most subjects, providing strong evidence that we are observing the same cell mosaic with both techniques. Although the modal TFI spacing of S#3 displays a small shift with respect to the NIRAF modal spacing, they appear relatively close. Cell density and spacing were estimated from the modal spacing extracted from these PSD graphs using the methods described in [25] and [26], respectively. The mean values for cell spacing and density computed from data of all subjects are very similar for both modalities and fall inside the uncertainty interval (see Table 3 and Bland Altman plots Figure 4). We were also able to identify the same cells in both TFI and NIRAF images (see yellow arrowheads in Appendix A), further strengthening the hypothesis that we are observing the same cellular structure.

We noticed that the clarity of the RPE mosaic varied according to the subjects. Figure 3 shows that subjects #1 and #2 generated a higher contrast image with distinct RPE cells. Subject #2 seems to display a sharper autofluorescence signal compared to the transscleral one, even though cells can be identified in both modalities. Subject #3 RPE cells are smaller than the other subjects in the same regions, rendering the distinction of cellular structure more difficult. Similar to subject #2, the autofluorescence signal looks clearer than the TFI signal that is modulated by the background low frequencies.

### 3.2. RPE Mosaic Variations with Time in Healthy Retina

The RPE mosaic at 10°T of subject #1 was imaged over three years at intervals going from minutes to months. We observed that the RPE mosaic seems stable when imaged minutes apart, suggestive of the same amount of pigmentation in the same RPE cells (see Figure 5). However, cell shape and the cell center level of intensity varied over months. In Figure 5, we show that although some cells remain the same, others show a very dark center in the T0 acquisition, showing a grayer cell center in the next acquisition 6 months and a year later. In addition to these changes inside the cell, some of these appear elongated in some dimensions at different times. For instance, in Figure 5, the green arrowheads identify a cell whose dark center is larger at T0 than at T0 + 6 m.

## 4. Discussion

We imaged the RPE cell mosaic for the first time using two different imaging modalities and identified many of the same cells in each modality. Although showing images generated from two different types of signal, we were able to observe with both systems the same cell morphology, the same modal spacing peak in the power spectrum densities of the images, and similar estimated cell density (Figure 3 and Figure 4). In addition, cell spacing and density values were also in accordance with RPE densities in these eccentricities [27]. We were able to validate the fact that we are observing the same retinal cellular structure identified as RPE in autofluorescence images in TFI images. Two interesting aspects of the TFI system are the fact that it provides a 7-time larger image field area (4° × 4°) than typical AOSLO images (1.5° × 1.5°) and that its acquisition time (6 s) is 5 times shorter than the minimum acquisition used in NIRAF AOSLO (30 s). In other words, if we compare the time needed for capturing the same retinal area, TFI is 35 times faster, which makes this system adapted to a clinical environment.

### 4.1. Origin of RPE Cell Contrast

Imaging the RPE layer using two modalities whose contrasts result from different light interactions with the tissue provides us additional information on these retinal cells, such as the level of pigmentation or fluorophore distribution, and may help us understand and determine more about the origin of the contrast in NIRAF, but also in TFI.

In the TFI system, we illuminate the retina through the sclera in a symmetrical manner, both light beams simultaneously shining through opposite sides of the pupil (see Figure 1). This symmetrical illumination of the same region implies an optical addition of the absorption terms of the detected signal and the cancellation of the phase terms. However, the retina has complex, highly scattering properties, and also, although the light beam enters at opposite sides of the pupil, we cannot positively describe how the region is being illuminated. For these reasons, although we hypothesize that absorption by the cell is responsible for most of the contrast, we cannot completely rule out a phase contribution, even if it is probably small.

We believe that, through absorption, the TFI modality removes the highly backscattering photoreceptor signal, and we can produce a contrasted RPE mosaic. Darker spots of TFI images correspond to regions with higher absorption, which are most likely due to higher densities of pigment granules such as melanolipofuscin [28,29]. In particular, it has been shown through histological images of RPE using laser scanning microscope (LSM) and structured light microscope (SIM) that lipofuscin granules are pushed towards the basolateral cell borders because of the nucleus, while there is a high content of melanolipofuscin granules at the center, which led to a larger region of hypoautofluorescence signal which do not exclusively represent the cell nucleus [28,30]. This is also observed in NIRAF imaging *in vivo*, where there is a large hypoautofluoresecent cell center (Figure 6). Similarly, in TFI images, some dark centers appear quite large (Figure 2), more than the typical size of a nucleus, which could also be due to the accumulation of these pigment granules around the nucleus leading to higher absorption and generating the hyposignal at the center of the RPE cells. This granule distribution is also supported by our observation in TFI images of the variation in intensity levels of the cell center, which could be attributed to variation in the number of granules inside the RPE cell leading to changes in absorption (see enlarged regions in both Figure 2 and Figure 6).

Finally, an interesting difference between NIRAF and TFI RPE images is the absence of vessel shadow on the TFI RPE mosaic. This is most likely because red blood cells in the vessels have absorbed the NIRAF light, the shorter wavelength of which is closer to their absorption spectrum. This hypothesis is further demonstrated in previous studies [10,21] where a shorter wavelength led to darker shadows of vessels on the RPE mosaic generated by AO autofluorescence imaging.

Unlike autofluorescence imaging in histological RPE tissue [32], in NIRAF imaging *in vivo*, we do not seem to distinguish the hypofluorescent/nonfluorescent gap between individual RPE cells, nor does it appear visible in TFI images either. This is most likely due to a lack of resolution as we are limited to a lateral resolution of approximately 2 μm. A potential limitation arising from this is the inability to distinguish between a multi-nucleated RPE cell, which has been highly documented in the literature [30,32,33,34], and two cells. Even though RPE multinucleation has not been associated with any failure of the RPE function yet, it has been noticed to increase with age and would thus be an interesting biomarker to obtain *in vivo* too, requiring further improvement of our imaging modalities [24].

### 4.2. RPE In Vivo Signal Variation with Eccentricity

Neither imaging modality shows a homogeneous RPE mosaic across eccentricities, with cells displaying a distinctive hyposignal center surrounded by a bright ring at around 10°T, which seems to become further mixed with other signals closer to the fovea. This observation in NIRAF is in accordance with autofluorescence imaging in histological RPE samples, where some authors noticed that perifoveal RPE cells (corresponding to eccentricities between 10° and 18° around the fovea) appear to have the highest autofluorescence signal compared to foveal and near-peripheral RPE cells [32]. This could be due to a higher lipofuscin granule load, which, as previously shown [28], is pushed towards the borders of the cell cytoplasm, generating these distinctive bright borders similarly to what is observed in Figure 2 NIRAF images. Other autofluorescence studies [10] have shown more contrasted RPE mosaics at larger eccentricities, particularly in the temporal retina. That study suggested that cell visibility is affected more by regional retinal characteristics than the age or ocular quality of the participants [10]. Interestingly, perifoveal locations in TFI images also display clearer RPE mosaics, suggesting a granule distribution that leads to more contrasted TFI images than in foveal regions. In addition, the larger size of the RPE cells in perifoveal locations compared to fovea [32] facilitates the resolution of individual cells at further eccentricities in both modalities. One concern raised in previous work developing RPE imaging was the possibility that there might be some signal from the rods at certain eccentricities which could be mistaken for RPE cells when they were forming a ring of single cells around the cones [15]. Our region of interest of 8°T–10°T corresponding to 2.5–3.1 mm is beyond eccentricities around 1.35 mm displaying these types of rings and rods [35,36], suggesting the observed cells are indeed RPE cells.

Another aspect influencing the RPE cell contrast in both modalities is believed to be a potential modulation of the NIRAF signal by the photoreceptors [10,11]. In Figure 2, bright small spots stand out from the background in NIRAF images close to the fovea, which has also been observed in previous *in vivo* autofluorescence studies of the RPE [10,11,13]. One hypothesis is attributing the autofluorescence in the pattern of cones to their waveguiding properties as the potential mechanism for this modulation. This could be the result either from the ingoing excitation light, which would lead to focused autofluorescence light at the tip of the outer segments, or on the way out, with bidirectional waveguides guiding the autofluorescence out of the eye through the cone. Although TFI image contrast is supposed to originate in absorption, it also seems to be mixed with brighter, smaller spots of the size of photoreceptors in foveal locations. This transscleral system is a full field modality, which, unlike confocal systems, has poor optical sectioning and detects light from all retinal layers. The transmission signal could thus be affected by the photoreceptors layer, leading to some modulation of the RPE mosaic. This trend in the RPE mosaic in subject #1 (Figure 2) was also observed in subjects #2 and #3, although with less strength in the NIRAF image in subject #3 (see Figure A1 and Figure A2 in Appendix B).

### 4.3. RPE Pigmentation Dynamic

We were able to detect a variation of the grayscale level inside the same RPE cells over certain periods of time in TFI images. We excluded changes in the image acquisition, such as subject head position in the system, by acquiring several image sequences at 10-min intervals and asking the subject to stand back in between. In particular, given the nonexistent optical sectioning of the TFI, we wanted to verify that the observed changes were not generated by variations from other regions of the retina, like the choriocapillary flow. Additionally, unlike the AOSLO, the TFI system has a full field acquisition, which prevents distortion artifacts in the average images, allowing us to interpret the observed changes as physiological variations. Furthermore, while minor fluctuations in AO correction may impact the quality of individual frames, they would not be expected to give rise to the observed variations, which are deemed physiological. As shown in Figure 5, RPE cells remain stable over these intervals. However, images acquired over a few months show, at several locations, changes in the apparent gray levels of the RPE cell (marked in Figure 5 by blue arrowheads), which even seems to modify the cell shape and size appearance (marked in Figure 5 by green arrowheads). The contrast variations are illustrated in Figure A3 through standard deviation images corresponding to each set of zooms depicted in Figure 5. The red spots indicate regions where there are notable differences in intensity between the time intervals. We observed that in zoom sets representing shorter time intervals, there are scattered small red spots, most likely due to noise. However, in zoom sets derived from images taken over several months, we observed larger circular spots, indicating changes in the intensity at the center of the RPE cells. The fact that these variations inside the RPE cells are not observed in images acquired between short timescale intervals (10 min) suggests that they are derived from slow physiological changes in the cell. One interpretation of these changes inside the RPE cells could be that these could be driven by pigment organelle motility. Thus, these variations in the cells hyposignal could imply a dynamic of melanin-containing complex granules inside the RPE cell, which has actually been long reported in the literature [37]. Furthermore, Feeney’s study [37] reveals the existence of a dynamic and complex interrelationship between the various components of the phagolysosomal system and the melanin granules in the RPE cytoplasm [37]. Although phagocytosis in the RPE occurs in a diurnal fashion and the entire photoreceptor outer segment population is turned over every 2 weeks [38], it does not entail that the melanin-containing granules follow that cycle or that enough changes in the amount of granules have occurred in that time to affect the contrast of the images. Similarly, other granules and structures are produced or absorbed by the RPE cell during this cycle, which could also affect the level of light absorption and, therefore, the contrast of the RPE mosaic in TFI images. Nevertheless, we observed for the first time in healthy subjects dynamic changes in the structure of the RPE mosaic at the level of single cells *in vivo*, which differs from the changes described in patients with retinopathologies such as age-related macular degeneration, where large clumps of pigment migrate over distances of several microns [21].

## 5. Conclusions

We successfully imaged individual RPE cells in the same living eyes using two different cellular resolution modalities. We show, for the first time, that the same cells could be imaged and overlaid 1:1 between modalities, cross-validating each of these tools for imaging of individual RPE cells. This validates that the observed mosaic is a cellular structure whose morphology and size match the RPE. We also show for the first time *in vivo* variability in the contrast and structure of individual healthy RPE cells, suggestive of pigment redistribution, which could allow us to extract biomarkers of healthy functioning RPE and thus better understand the role of pigment in the RPE. However, it must be noted that in this study, we compared images from the TFI system to NIRAF images from previous work on the same subjects [11] generated several years ago, representing a considerable time gap between the two datasets. Given that we observed dynamic changes within the RPE layer using the TFI system over intervals of months, one limitation of this study was the fact that we could not compare intracellular features between the two imaging modalities despite observing the same cells. Therefore, in the future, we will expand our study cohort to include more subjects and acquire images during the same imaging session for both modalities, allowing for a more robust analysis of intracellular comparisons between NIRAF and TFI imaging modalities. Finally, the addition of TFI and NIRAF AOSLO to future multi-modal imaging studies may provide complementary information about RPE structure and may help to better understand RPE pigment redistribution in different pathologies and with age. In particular, due to its longitudinal monitoring capability, we will introduce the TFI system into clinical protocols to follow the progression of diseases affecting the RPE layer, such as AMD. The TFI system is particularly adapted to a clinical setting because of its short acquisition time and large field of view.

## Figures and Tables

**Figure 1 diagnostics-14-00768-f001:**
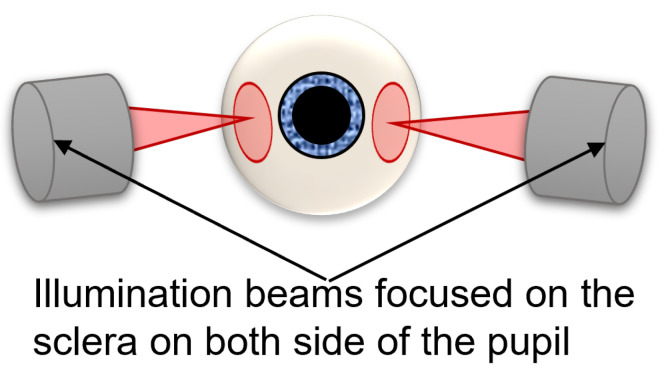
Schematic of the transscleral illumination of the TFI system.

**Figure 2 diagnostics-14-00768-f002:**
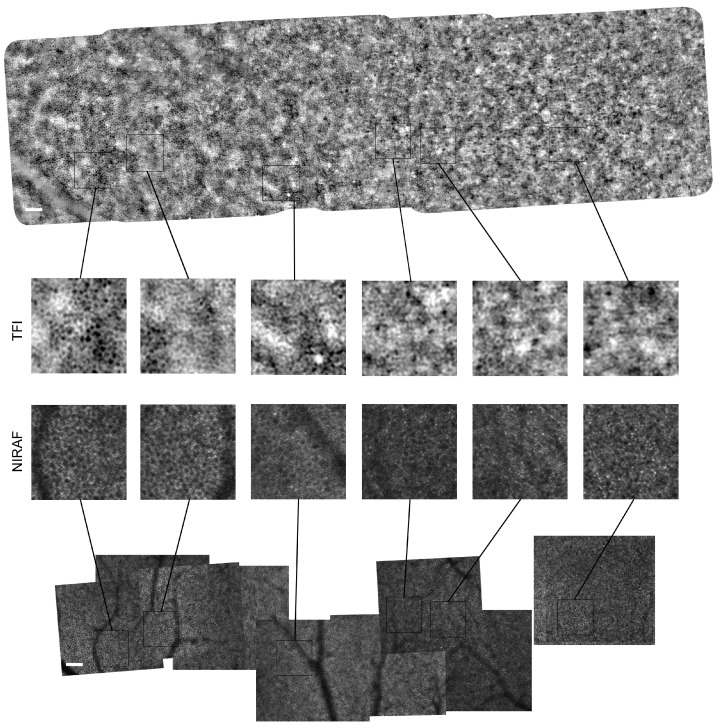
Montages of RPE layer images acquired with the (**top**) TFI and (**bottom**) NIRAF modality from the fovea (**right**) to 10°T (**left**) on subject #1. Enlarged regions are compared at various eccentricities. Scale bar is 100 μm. Appendix B Figure A1 and Figure A2 show montages for subjects #2 and #3.

**Figure 3 diagnostics-14-00768-f003:**
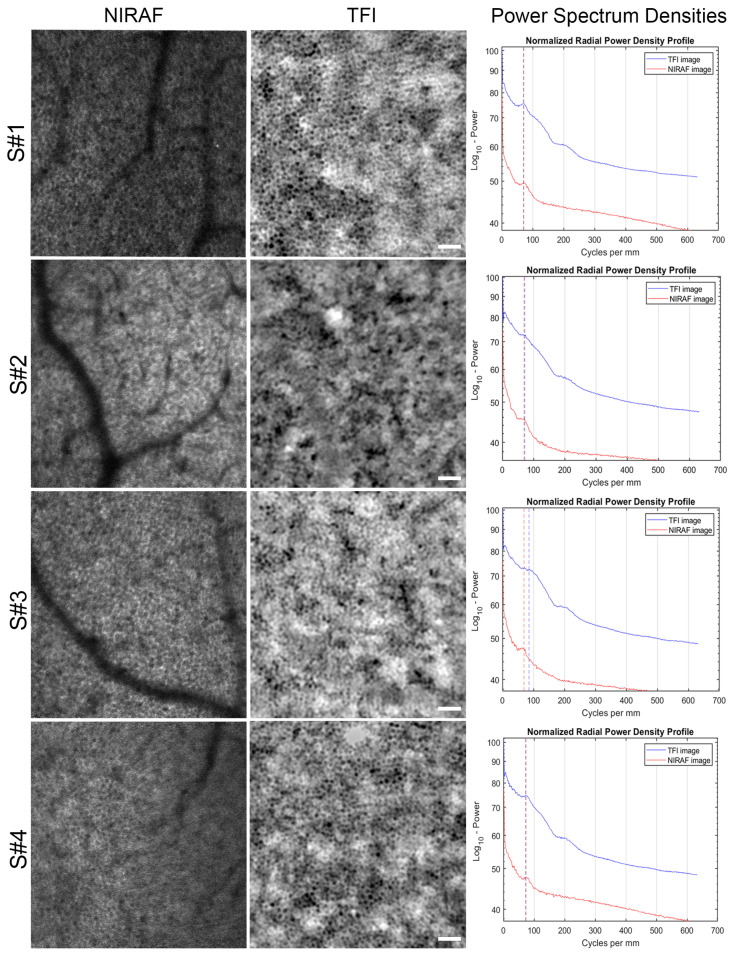
Comparison of RPE images taken with NIRAF and TFI modalities on the same region (10°T). The power spectrum densities were computed on each image and superimposed on the last column. Dashed lines highlight the peaks corresponding to the modal frequency of the RPE cell spacing, in blue for TFI and red for NIRAF. Scale bars are 50 μm long. Appendix A shows the superposition of TFI and NIRAF images for all subjects.

**Figure 4 diagnostics-14-00768-f004:**
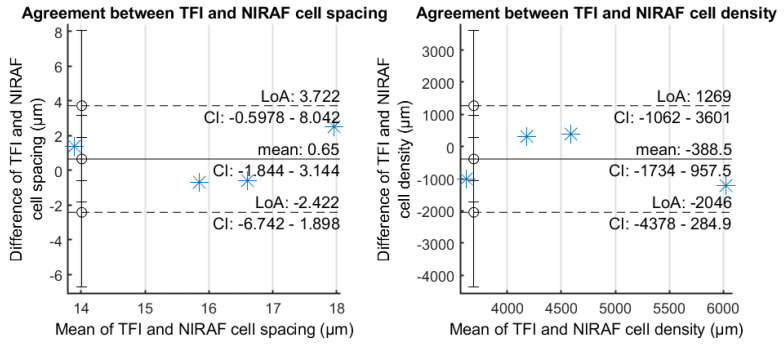
Bland Altman plots comparing TFI and NIRAF cell spacing and density. The plots show the agreement among the two different systems as measures fall inside the horizontal lines, i.e., inside the limits of agreement.

**Figure 5 diagnostics-14-00768-f005:**
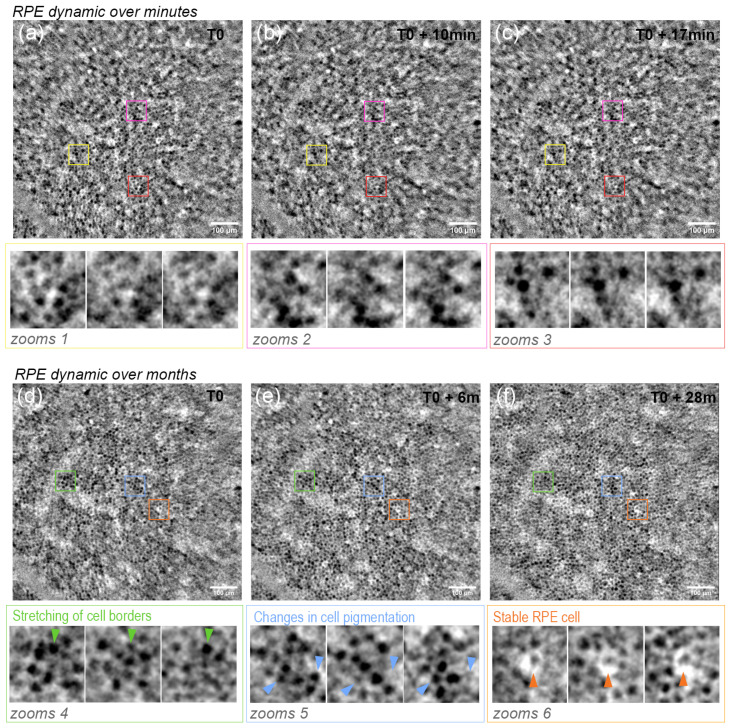
Longitudinal imaging subject #1 RPE mosaic (**a**–**c**) over minutes and (**d**–**f**) months. Three zoomed regions for short-term intervals (zoom 1–3 for (**a**–**c**)) and for long-term intervals (zoom 4–6)) show details of single RPE cells. Scale bars are 50 μm.

**Figure 6 diagnostics-14-00768-f006:**
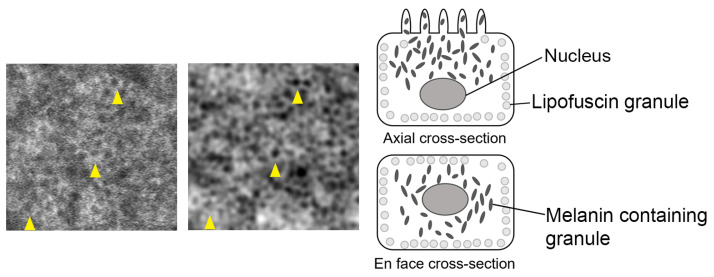
(**Left**) Enlarged region of subject #4 in Figure 3 showing same cells in NIRAF and TFI contrasts with yellow arrowheads showing some examples. (**Right**) Schematics of melanin-containing granules such as melanolipofuscin granules (dark circles) and lipofuscin (light circles) granules simplified arrangement inside an RPE cell suggested by histology findings in [28,29,30,31].

**Table 1 diagnostics-14-00768-t001:** Paris and Pittsburgh NIRAF AOSLO parameters.

	Paris NIRAF	Pittsburgh NIRAF
**Wavelength (nm)**	757	720
**Field of view (°)**	2	1.5
**Acquisition time (s)**	90	30

**Table 2 diagnostics-14-00768-t002:** Subjects image acquisition details for each modality.

Subjects	AO System	Regions Imaged	Year of Image Acquisition	Age of the Subject during Acquisition
Subject #1	TFI	Fovea-10°T	2023	31 yrs
		10°T	2021 to 2023	29–31 yrs
	Paris NIRAF	Fovea-10°T	2017	25 yrs
Subject #2	TFI	Fovea-10°T	2023	44 yrs
	Pitt NIRAF	8°–10°T	2022	43 yrs
Subject #3	TFI	Fovea-10°T	2023	25 yrs
	Pitt NIRAF	8°T	2023	25 yrs
Subject #4	TFI	Fovea-10°T	2023	28 yrs
	Paris NIRAF	Fovea-10°T	2017	22 yrs

**Table 3 diagnostics-14-00768-t003:** RPE cell spacing and density extracted from the Power Spectral Density of TFI and NIRAF images for all subjects.

	S#1		S#2		S#3		S#4		Mean +/− SD	
	TFI	NIRAF	TFI	NIRAF	TFI	NIRAF	TFI	NIRAF	TFI	NIRAF
Modal spacing (cycles/deg)	19.8	20.6	20.1	17.5	25.5	23.0	20.7	21.6	21.5 +/− 2.3	20.7 +/− 2
Cell spacing (μm)	16.9	16.3	16.7	19.2	13.2	14.6	16.2	15.5	15.7 +/− 1.5	16.4 +/− 1.7
Cell density (cells/mm)	4027	4327	4133	3117	6629	5411	4397	4777	4797 +/− 1066	4407 +/− 839

## Data Availability

Data presented in this study is not available for privacy reasons. Deidentified data could be provided upon reasonable request by the corresponding author.

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
