# Peer review of "Comparison between Two Adaptive Optics Methods for Imaging of Individual Retinal Pigmented Epithelial Cells"

_diagnostics, 2024, doi:10.3390/diagnostics14070768_

Round 1

Reviewer 1 Report

Comments and Suggestions for Authors

The paper describes measurements on healthy subjects with NIRAF and TFI imaging, looking specifically at the RPE cell mosaic. The images show agreement in the cone mosaic between the two modalities, quantified with normalized radial power density profile and by comparing the images. Differences are discussed.

Altogether the manuscript is well written with pretty retinal images and a thorough analysis of the data, including comparisons with the literature.

Please comment on the following concerns:

a) In the video, there is occasionally a small shift in cells from one frame with one modality to the other with the other modality. You comment on this, stating that the shift is small. What is causing the shifts?

b) “As shown in Fig.5, RPE cells remains stable over these intervals.” Looking at Fig. 5, there is some variation. Remaining stable is perhaps subjective. Can you quantify these changes, and compare them to the changes over longer periods of time (lower set of images)? You may want to add letters to the various panels, to make it easier to distinguish between short-term images and long-term images.

c) There is a (lack of) relationship between the NIRAF signal and the TFI signal. Is there any value in the ratio of the two in the center of cells and edges of cells, and how the ratio varies from person to person, and in a person over time? You have access to these data; it seems a waste not to have a closer look at it!

d) “Figure 4. Bland Altman plots comparing TFI and NIRAF cell spacing and density.” -> Please comment in one sentence what can be concluded from these plots in the caption.

e) Figure 5, stretching of cell borders. Nice observation – how can you conclude that this is cell stretching and not a result of aberrations / imaging. How consistent are images on the same day? Just a comment on this may be sufficient.

f) “For these reasons, although we hypothesize that cell’s absorption is responsible for a major part of the contrast, we cannot completely rule out a phase contribution, even if probably small.” -> This needs to be introduced better. What is a phase contribution, where does it originate from.

Minor comments:

“The Retinal Pigment Epithelium (RPE) plays” -> The retinal pigment epithelium (RPE) plays

“although it is not yet well understood” -> what specifically is not well understood?

“an assessment the state of the layer” -> an assessment of the state of the layer

“Optical Coherence Tomography (OCT) generates cross-sections of the” -> Optical coherence tomography (OCT)…

“Transscleral optical phase imaging (TOPI)” -> transscleral

“prototype named Transscleral Flood Illumination (TFI) camera” -> prototype named transscleral flood

illumination (TFI) camera

“Both transscleral systems exploits” -> exploit

“healthy RPE cells obtaiend” => obtained

“Finally, all subjects were imaged on the TFI system in 2023, and additionnally images were acquired on subject 1 from 2021 until 2023.”

“ensuring the laser power were below the maximum MPE” -> was

“a custom made matlab software” -> a custom made mMatlab software

“Scale bas is 50 μm.” -> Scale bars are 50 μm long

“what seems small cells in an irregular arrangement” -> seem

“two different type of signal” -> two different types of signal

“of RPE using Laser Scanning Microscope (LSM) and Structured Light Microscope (SIM)” -> of RPE using laser scanning microscopy (LSM) and structured light microscopy (SIM)

“(see zooms in Fig.2,6).” -> rephrase?

“where shorter wavelength led to darker” -> a shorter

“as we are limited to around 2 μm” -> …as we are limited to a lateral resolution of approximately 2 μm … Is this FWHM or 1/e2?

“Eventhough RPE multinucleation hasn’t” -> Even though RPE multinucleation has not…

“Our region of interest of 8°T-10°T correspond to 2.5-3.1 mm is beyond” -> corresponding

“around 1.35mm displaying” -> 1.35 mm

“…changes weren’t generated by variations from other regions of the retina like the choriocapillary flow.v …” -> changes were not generated by variations from other regions of the retina like the choriocapillary flow.

“As shown in Fig.5, RPE cells remains stable over these intervals.” -> remain

Comments on the Quality of English Language

Found a few problems - these were addressed in my review report.

Author Response

We thank the reviewer for their suggestions. Please see the attached word document for the detailed response.

Reviewer 2 Report

Comments and Suggestions for Authors

Dear Authors,

I wish to submit my review for the paper "Comparison between two adaptive optics methods for imaging of individual retinal pigmented epithelial cells."

The authors performed an interesting research article comparing the contrast, image quality, and morphometric properties of RPE using near-infrared autofluorescence adaptive optics scanning laser ophthalmoscopy (NIRAF AOSLO) and trans-scleral flood imaging (TFI). The Authors provided novel research with interesting findings and should be commended for their work. Despite the novel and interesting topic, I have various concerns. 

Specifically, the abstract is concise and unstructured and does not summarize the research findings; therefore, it requires extensive proofreading.

Methods: The Authors enrolled only four patients without specifically reporting the inclusion/exclusion criteria. A considerable time gap between the two evaluations could have biased the results and should be considered a study limitation. Demographic data should be fully reported.

Discussion: The discussion should focus on the possible future clinical perspective of the presented findings. Finally, I suggest reporting the study’s limitations.

Comments on the Quality of English Language

 Minor editing of English language required

Author Response

We thank the reviewer for their suggestions and corrections. Please see the attached word document for the detailed response.

Reviewer 3 Report

Comments and Suggestions for Authors

The authors compared images of the healthy RPE cells from same objects obtained with the transscleral flood illumination (TFI) and with near infrared autofluorescence (NIRAF) images from AOSLO. The results are solid, and the images are clear and self-explanatory. Minor edits are listed as below:

How long does it take to obtain one image on average using TFI compared to NIRAF images from AOSLO?

Following the above question, authors should mention in the discussion their opinion on if there will be a broader application of TFI.

Line 60, it should be ‘obtained’.

Line 181, it should be ‘additionally’.

Line 294-295, ‘that’ was repeated.

Comments on the Quality of English Language

English is OK.

Author Response

(The authors gave the same response as above.)
